# Ferroelectret Polypropylene Foam-Based Piezoelectric Energy Harvester for Different Seismic Mass Conditions

Chandana Ravikumar * and Vytautas Markevicius

Department of Electronics Engineering, Faculty of Electrical and Electronics Engineering, Kaunas University of Technology, LT-51365, student street 50-438, 44249 Kaunas, Lithuania; vytautas.markevicius@ktu.edu
* Correspondence: chandana.ravikumar@ktu.edu

**Abstract:** Energy harvesting technologies and material science has made it possible to tap into the abundant amount of surrounding vibrational energy to efficiently convert it into useable energy providing power to portable electronics and IoT devices. Recent investigations show that the piezo-electric effect is created in cellular polymers called ferroelectrets. These cellular-compliant polymers with polarized pores have a piezoelectric response to generate electrical energy when subjected to mechanical strain or surrounding vibration. It is found that there is a significant difference between ferroelectret polarized cellular polypropylene foam and traditional piezoelectric polymers such as polyvinylidene fluoride (PVDF). The former has approximately ten times higher piezoelectric co-efficient than the latter. This means that with an acceleration of 9.81 m/s$^2$ force on this material, ferroelectrets generate up to 39 ($\mu$W/g/mm$^3$) power output. Designing a polypropylene-based piezoelectric energy harvester based on the d33 mode of vibration can be challenging due to several factors, as it requires balancing multiple factors such as mechanical stability, piezoelectric response, circuit topology, electrode size, spacing, placement relative to the piezoelectric material, and so on. This paper proposes the preliminary experimental investigation of ferroelectret cellular polypropy-lene foam in harvesting performance. Suggestions of different approaches for the structural design of energy harvesters are provided. The vibration-dependent response and generated output are examined concerning pulse or sinusoidal input excitation. The voltage generated for both excita-tions is compared and suggestions are provided regarding the suitable kind of excitation for the chosen ferroelectret material. Finally, conclusions and prospects for ferroelectret materials used in energy-harvesting applications are given.

**Keywords:** cellular polypropylene foam; ferroelectret; frequency response; material for energy; piezoelectricity; power density; self-powered electronics



## 1. Introduction

Abundant mechanical vibration is available in our immediate surroundings, piezo-electric energy harvesting has indeed been an everlasting source of excitation from the movement of cars, trains, airplanes, bikes, skis, the rotor of mechanical machine-like grass mowers, stone-cutting machines, funiculars, etc., [1–6]. All objects in our surrounding environment vibrate at different frequencies when they are hit, struck, plucked, strummed, or somehow disturbed. Commonly, vibration of objects produces complex waves with a set of multiple frequencies namely, harmonics that have no simple mathematical relation-ship between them. With these vibrations differing from each order concerning frequency distribution and magnitude, it is a challenging task to design particular energy harvesters suitable for each case to couple with the different sources. Fundamentally good coupling between the vibrating source and the energy harvester amounts to a higher energy transfer from the environment to the transducer [2,7]. This work is dedicated to the study of the influence of vibration waveform on the performance of the piezoelectric vibration energy harvester.

Piezoelectric materials could also be considered energy materials because of their ability to produce electricity on the application of strain or mechanical vibration. Recent investigations reveal that porous non-polar polymer foams that are used in disposable cups can exhibit behavior resembling piezoelectricity under high electric fields and are classified as ferroelectret materials [8,9]. The internally charged voids of cellular polymer foams show some micro discharges that convert them into ferroelectrets after the expansion process. After the procedure of electrical poling and surface metallization, the ferroelectret material is found to possess large piezoelectric d33 coefficients (charge per unit force) up to 300 pC/N [9,10]. The first ferroelectret polymer cellular polypropylene foam has achieved at least ten times higher piezoelectric coefficients compared to conventional polyvinylidene difluoride (PVDF) piezoelectric polymer that measures d33 value approximately 20–30 pC/N [11,12]. Even though lead zirconate titanate (PZT) (an inorganic piezoelectric material) is the dominating material for energy harvesting applications due to its performance and mechanical stability, it is also considered a toxic substance because of the lead constituent [13]. In addition to this, piezoelectric ceramics are stiffer, and more brittle compared to polymer ferroelectret materials [14,15], rendering them unsuitable for certain applications. In environmental monitoring systems, autonomous vehicles, wearable electronics, and other IoT applications the electronic circuitry is present at remote places where the accessibility to the electrical power supply is minimum or the possibility to replace or recharge the batteries may be too expensive or simply inconvenient. For instance, one of the sensors embedded in the road is a temperature sensor that is used for a simple application to know when the road becomes cold, and salt should be thrown on the road to avoid vehicle accidents on slippery icy roads. Although such a sensor could be placed on a road five years ago, the maintenance of the sensor would be a challenging issue because of the need to replace the batteries systematically. The application loses its feasibility when a road highway is required to be closed down because of the exhaustion of the batteries supplying power to the sensors. Energy harvesting from the surrounding environment to supply power to the sensors in such applications is a promising way to make the entire application realistic. A vibration-based energy harvester that generates electricity by converting the mechanical vibrations of vehicles passing to usable power can become a solution for powering the electronic circuitry under the road.

This study of vibration-based energy harvesters comes under the board umbrella of mechanical conversion devices. The four main energy conversion mechanisms used for vibration-based energy harvesting are electromagnetic, electrostatic, piezoelectric, and triboelectric. The literature review shows certain shortcomings of electromagnetic energy harvesters, because of the use of permanent magnets to produce power from induced current in magnetic fields, the size of the harvester is comparatively bigger and hence there are problems of mechanical complexity in this mechanism [16,17]. In the electrostatic and triboelectric mechanism, it is hard to ensure continuous and secure production of energy, so this mechanism is limited to only a few applications due to its low current [18–20]. In the piezoelectric conversion mechanism, the needed voltage is delivered immediately not requiring a separate voltage source as in the case of electrostatic conversion. Also, for device sizes less than 1 cm$^3$, piezoelectric mechanism shows higher energy conversion capacity compared to electromagnetic harvesters [5,21]. A thorough literature review suggests that there are far more advantages in piezoelectric-based vibration energy harvesters (PVEH) compared to their other counterparts, mainly because of the simple architectures and bigger energy density along with the additional features of ease in the scalability of piezoelectric and ferroelectret material in micro and nanoscale devices [14–16].

This study identifies the challenges in designing a polypropylene-based piezoelectric energy harvester based on the d33 mode of vibration, it proposes an experimental setup to investigate the energy harvesting performance of ferroelectret polypropylene and compares it with the literature. This paper also compares the response of the ferroelectret polypropylene foam to pulse and sinusoidal signal excitation types that helps predict the energy harvesting performance of the material in real-life vibration environments. Factors

assessing the electromechanical response of the ferroelectret materials are discussed. A brief explanation of the production procedures for ferroelectret polymer structures for energy harvesting is provided. The difference between piezoelectric and ferroelectret materials is elaborated. The different ferroelectret materials that are available for harvesting applications are listed. Future recommendations and ideas are provided.

## 2. Related Works

The performance of energy harvester systems largely depends on the sensitivity of the piezoelectric material, and hence more is being done to find and investigate more materials. Similar to polypropylene other ferroelectret materials being developed are mainly polypropyleneIrradiation-crosslinked polypropylene (IXPP), fluorinated ethylene propylene(FEP), and Polytetrafluoroethylene (PTFE).

It is also worth noting that though fluorocarbons show some superior properties, their properties are bound by expensive and complicated processing methods [22–24]. Polypropylene is rather cheaper and is a thriving thermoplastic that possesses equally promising energy-generating properties. Foam is a composite of gas and polymer, while PVDF is a pure polymer. This means that PP ferroelectret foam has lower density and lower material cost than PVDF. The major drawback of low thermal stability of PP foam can also be overcome by the recent developments in this regard, the charge storage and thermal stability of isotactic polypropylene have been improved by the addition of special nucleating agents at an optimal concentration [25]. So, both PP foam and fluoropolymers (PTFE/FEP) have some advantages and disadvantages, having a trade-off between high piezoelectric constant and moderate thermal stability and flat frequency response, multilayer stacked PP foams have a higher stand over PTFE or FEP. Therefore, one can confidently choose to work with ferroelectret PP foam in low-level vibration energy harvesting applications as a lead-free and extremely compliant alternative to conventional piezoelectric materials.

PP foams have been produced using thermoplastic and nonpolar polymers such as polyolefin (PO), cyclo-olefin polymers (COP), and polyurethane (PU) [26,27]. Such non-porous polymers behave ferroelectrically when exposed to a strong electric field, it is internally charged by microplasma discharges within the voids of the polymer and when an external electric field is applied, the polarity of these engineered dipoles switches alternately showing piezo and pyroelectric properties. When a gas, such as air, is within a macro-sized pore space (usually >1 m) of a ferroelectret, the piezoelectrically active polymer foam becomes vulnerable to electrical breakdown in the process of a high electric field by a corona poling.

Figure 1 shows the fabrication process by which internally charged cellular polymers provide a novel class of ferroelectret materials such as polypropylene (PP) films. Authors in reference [15] purchased a multilayer PP film (Treofan film EUH75) and at a temperature of 100 °C for 3 h pressure of 2 MPa is applied. This increased the film thickness from ≈75 μm to ≈150 μm, visible in Figure 1B,C. The material is placed 5 cm below the corona needle and applied with a high voltage of −16 V for 3 min. The voids in the material are ionized and become polar in nature. The upper and lower pore surfaces of the polymer are the place where the microplasma discharge is deposited [28,29]. This is the outcome of the breakdown of air in the pore space as seen in Figure 1. When the polarized pores are produced, an electrode ferroelectret will draw surface charges, and if the polarization of the pores is altered mechanically, there is a redistribution of the surface under stress or a temperature change, the flow of electricity, and a charge. The samples are coated with silver on both sides using a magnetron sputtering system. With this process, it is evident that more charged voids yield a higher $d_{33}$ coefficient value, which in turn means that higher output is generated. Consequently, ferroelectrets are both pyroelectric and iezoelectric, and such characteristics have just sparked curiosity about energy harvesting applications.

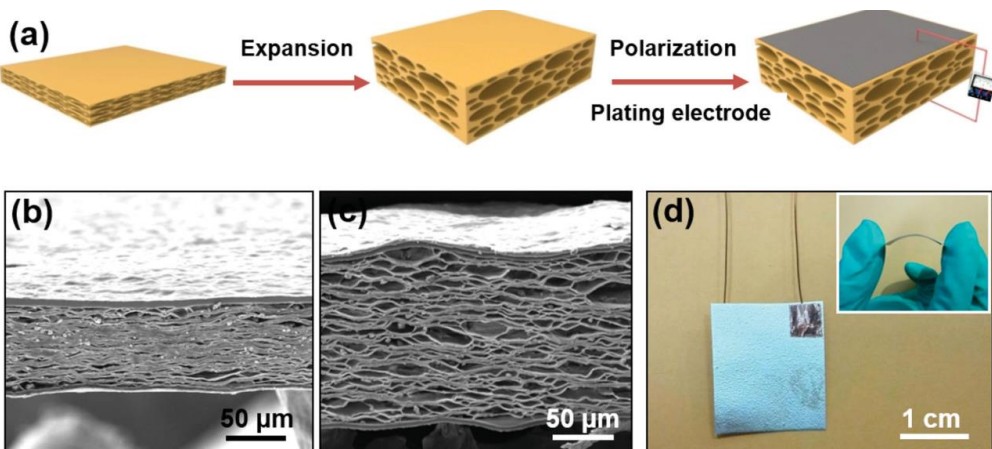

**Figure 1.** (**a**) Schematic of the fabrication process. Cross-section view of cellular polypropylene (**b**) before and (**c**) after the expansion process. (**d**) Image of final poled and electrode device [15].

Designing a polypropylene-based piezoelectric energy harvester based on the d33 mode of vibration can be challenging due to several factors. The harvester's structural design should be optimized for the d33 mode of vibration, which involves applying mechanical stress in the direction perpendicular to the polarization direction of the material. Achieving the optimal design can be challenging, as it requires balancing multiple factors such as mechanical stability, piezoelectric response, and manufacturability [30,31]. By applying different weights of seismic masses, it is possible to adjust the mechanical damping of the system and explore the optimal conditions for energy harvesting. A higher weight of seismic mass can increase the mechanical damping and reduce the mechanical deformation of the ferroelectret material, which can lower the output power and efficiency. On the other hand, a lower weight of seismic mass can reduce mechanical damping and allow for a higher degree of mechanical deformation, which can increase output power and efficiency [14,32]. Hence, one of the aims of this work is to study the effects of three different values of seismic masses on the frequency response of the energy harvester.

Also, studies confirm that the behavior of a ferroelectret material is different for different kinds of waveforms of input vibration frequency due to its inherent material properties and the response of the material to external stimuli. The ferroelectric material exhibits different electrical responses when subjected to different kinds of waveforms, which can affect its energy conversion efficiency [33,34]. This study will provide information about the influence of the applied perpendicular force and waveform of input excitation on the overall performance of the harvester in terms of resonant frequency, output voltage, and power generated by the harvester.

Overall, designing a polypropylene-based piezoelectric energy harvester based on the d33 mode of vibration requires careful consideration of multiple factors to achieve optimal performance. Since the discovery of this material just 20 years ago [8] there is an enormous potential for additional progression of cellular polymer-based piezoelectric energy harvesters. Hence, there is still a need to find the most appropriate approach for assessing polypropylene (PP) foam through experimental investigation of the material in an energy harvesting scenario that can allow one to prepare a suitable experimental setup for the investigation of the material to study the frequency response of the material under different perpendicular force conditions and behavior of the material to different kinds of waveforms of input vibration frequency i.e., sinusoidal or pulse input. A comparision of exisiting works based on the d33 coeffient measured and the power density generated is of polypropylene foam is shown in the Table 1 below.

**Table 1.** Summary of Recent Works of Ferroelectrets Used for Energy Harvesting Applications.

| No | Material | d33 (pC/N) | Area (mm$^2$) | Power Density (µW/g/mm$^2$) | Source |
|----|----------|------------|---------------|----------------------------|--------|
| 1 | PP (1 layer) | 300 | 3000 | 0.34 | [15] |
| 2 | | 300 | 8400 | 0.41 | [34] |
| 3 | | 250 | - | 0.05 | [17] |
| 4 | PP (10 layers) | 300 | 3000 | 3.3 | [15] |
| 5 | | 220 | | 7.20 | [17] |
| 6 | PP (1 layer) | 300 | 7 | 39 | Present work |

Calculated data based on relative parameters reported in the reference.

## 3. Methodology

*Experiment Setup*

Since PP foam is dominant in the d33 mode of vibration, the force should be applied vertically on the material unlike in PVDF film where force is applied horizontally by the pull from the tip mass. For PP foam the force should act on the material as a pressing force in a hammering motion. After experimenting with a few setup structures finally, a stable framework was made as shown in Figure 2b. There is a metal rod that works as a hammer and the hammering force is adjusted by using a suitable number of metal washers screwed to the metal rod. The Top and bottom electrodes coated with aluminium are added to the structure so that the test sample can be sandwiched between them as shown in Figure 2a.

Seismic masses are used in the d33 mode of piezoelectric energy generation experiments to apply mechanical stress to the piezoelectric material in a direction perpendicular to the polarization direction. This mechanical stress generates an electrical charge on the surface of the piezoelectric material due to the d33 piezoelectric effect, which can be harvested as electrical energy. The seismic mass is typically a heavy metal or ceramic material that is attached to the piezoelectric material using a mechanical coupling, in our case we use a metal seismic mass of a minimum of 50 g and a maximum of 150 g. The average seismic mass weight used in piezoelectric energy harvester setups in d33 mode can vary depending on the specific application and the properties of the piezoelectric material being used. However, in general, seismic masses used in d33 mode piezoelectric energy harvesting experiments can range from a few grams to several kilograms [35,36]. When the seismic mass is subjected to an external force, it applies mechanical stress to the piezoelectric material, causing it to deform and generate an electrical charge on its surface. In the experiment setup, an electromagnetic shaker is used to provide the excitation force. The use of seismic masses is essential for d33 mode piezoelectric energy generation experiments because it allows the piezoelectric material to be mechanically stressed in a direction perpendicular to its polarization direction, which maximizes its d33 piezoelectric response. Without the seismic mass, it would be difficult to generate a strong enough mechanical stress in the desired direction to achieve a significant energy output.

A typical electromechanical system of a vibration energy harvesting system is shown in Figure 3a. The voltage generated by an energy harvester has to be rectified and then stored in a supercapacitor or a battery for further use. The flow chart in Figure 3b illustrates the design of the experimental setup, consisting of a signal generator (RIGOL DG5251) and a power amplifier to provide a signal to the exciter. The amplifier is typically connected to the output of the signal generator and acts as a voltage amplifier, boosting the amplitude of the input signal to a level that is sufficient to drive the ferroelectret material effectively. The use of an amplifier with a signal generator can improve the sensitivity and performance of the energy harvester, enabling it to generate more electrical power and operate more efficiently. Moreover, the use of a stable and well-defined input signal can help to reduce the noise and variability in the electrical output of the ferroelectret material, improving the reliability and accuracy of the energy harvester.

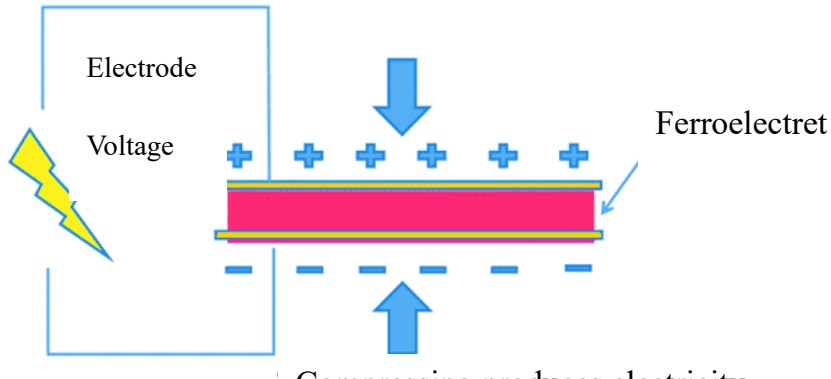

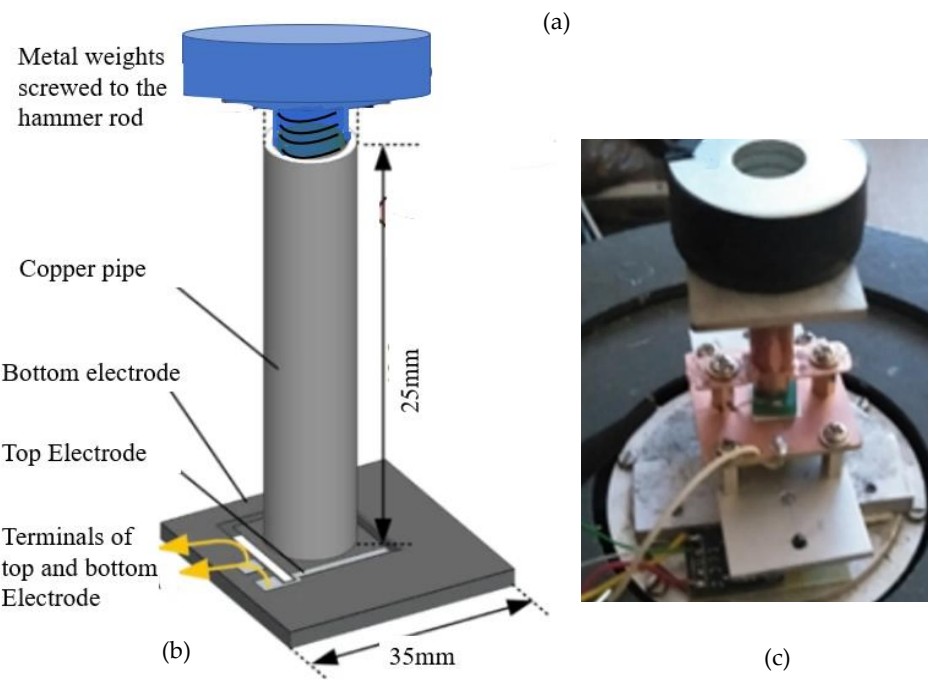

**Figure 2.** (**a**) Piezoelectric effect in Ferroelectrets (**b**) Model of the mechanical fixture used in experiment setup (**c**) Photo of mechanical fixture.

An electrodynamic shaker (MESSELEKTRONIK) is used to excite the sample placed within the mechanical fixture fixed on the shaker. A function generator (RIGOL DG5251) and a power amplifier supply the excitation signal to the shaker. A digital oscilloscope (RIGOL DS1302CA) is used to measure the voltage generated by the ferroelectret material. The ferroelectret sample is fixed on the exciter using the mechanical fixture as shown in Figure 2b. The mechanical fixture is vertically mounted on the exciter where the ferroelectret film is fixed at both ends of the supporting structure.

The ferroelectret PP foams were commercially purchased from Emfit Ltd. (Jyväskylä, Finland). The commercial samples come in sheets of size 230 mm × 210 mm, with a thickness of 70 μm. Their piezoelectric charge constant d33 was mentioned to be 300 pC/N. The PP foams were further cut into testing samples of size 10 mm × 10 mm. Since the sample size was as small as just 10 mm$^2$, a sharp blade was used as the cutting tool, however, one must be cautious to control the cutting parameters, such as the cutting speed, angle, and force, to minimize the introduction of mechanical stresses and strains into the material. Additionally, the cut samples were carefully characterized and tested to ensure that they exhibit the desired properties and meet the requirements of the energy harvester application.

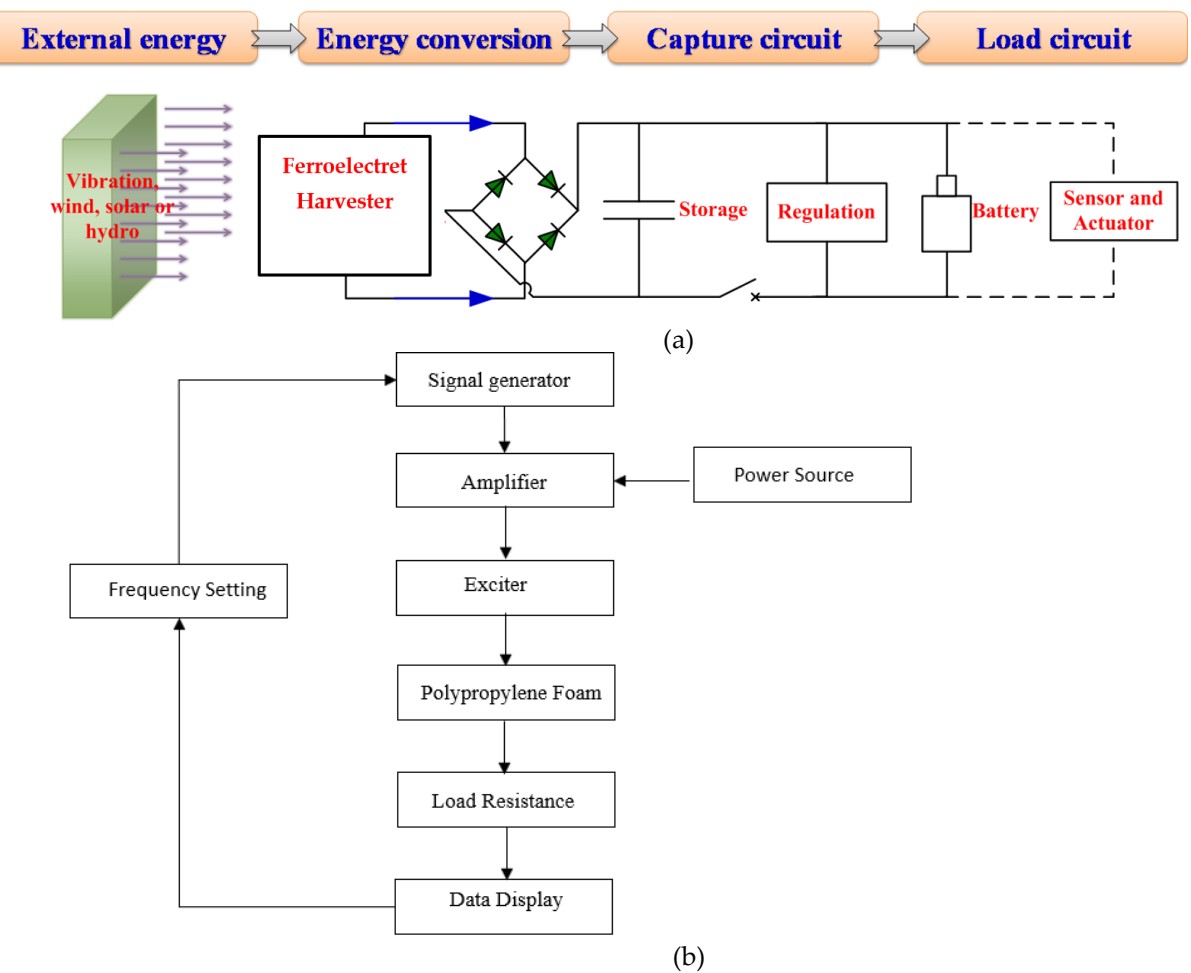

**Figure 3.** (**a**) Schematic of typical piezoelectric energy harvesting system (**b**) Block diagram of the Experiment setup.

## 4. Selection of Input Excitation Waveform

The behaviour of a piezoelectric material is different for various kinds of waveforms of input vibration frequency due to its inherent material properties and the response of the material to external stimuli. The input excitation waveform shape is important in piezoelectric energy harvesting because it directly affects the output power and efficiency of the energy harvester. Different waveform shapes can excite different modes of vibration in the piezoelectric material and affect the distribution and amplitude of the electrical output [37,38].

A sinusoidal excitation waveform can excite the fundamental resonance mode of the piezoelectric material and produce a higher output power compared to other waveform shapes, such as triangular or square waves, which can excite multiple resonance modes and produce lower output power. Additionally, the waveform shape can affect the efficiency of the energy harvester by influencing the energy losses and damping in the system. For example, a waveform with a higher degree of symmetry can produce lower damping and losses in the system compared to a waveform with more abrupt changes, which can lead to higher energy conversion efficiency [39]. Therefore, selecting a suitable waveform shape is important to maximize the output power and efficiency of the piezoelectric energy harvester. The choice of waveform shape should take into account the specific requirements and constraints of the target application, including the type and frequency of the input vibration, the available power electronics, and the desired level of power output and efficiency.

In the surrounding environment, different mechanical vibrations occur mainly in outdoor mechanical machines such as grass mowers, stone-cutting machines, automobiles,

etc. The occurrence of these vibrations can be either periodic or random in nature. Mostly environment vibrational patterns are aperiodic and in terms of frequency content, these vibrations can only be defined as having their energy distributed continuously over some range or "band" of frequencies. For the sake of experimentation, aperiodic vibrations are presented as pulse vibrations at a specific duty cycle in this study. The sample is energized with a pulse input signal from a signal generator having a 50% duty cycle. However periodic waveforms can take the shape of sine, triangle, square and sawtooth, but since the sinusoidal waveform is the fundamental waveform, one can approximate the other periodic waveforms with the addition of sine waves at the appropriate harmonics, and the appropriate amplitude levels. Since majorly in the environment, we see vibrations having a damped sinusoidal or pulse waveform, the study presented in this work compares the voltage generated by ferroelectret PP foam for the same amplitude of sinusoidal and pulse excitation waveforms.

The PP sample is energized with a pulse input signal from a signal generator having a 50% duty cycle. The peak-to-peak input voltage Vc given to the exciter coil is measured by the yellow channel of the oscilloscope (Figure 4a). The voltage given to the exciter coils is indicated by the yellow channel in the oscilloscope, the acceleration on the diaphragm of the exciter is shown by the green channel, and the corresponding raw voltage generated by the sample is shown in the blue channel. When an acceleration of 0.4 g acts on the PP sample with 50 g of pressing weight, for the excitation waveform of a pulse signal, the raw output peak-to-peak voltage generated by the sample is 0.640 V. Similarly, for the same acceleration but for sinusoidal excitation waveform PP sample generates 1.2 V as listed in Table below. The procedure was conducted five times and an average of the results is shown.

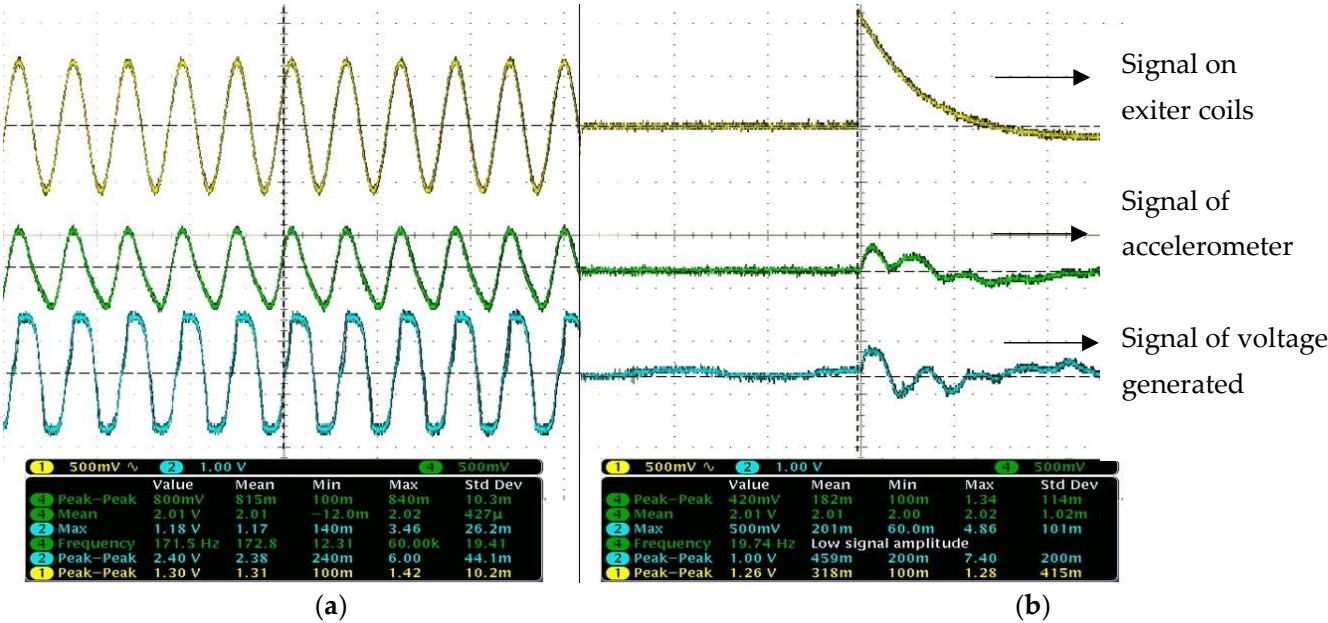

**Figure 4.** Oscilloscope screenshot display where the Yellow channel shows voltage on exciter coils, the green channel shows acceleration measured on the exciter diaphragm, and the blue channel is the voltage generated for (**a**) pulse input of 0.4 g (**b**) Sinusoidal input of 0.4 g.

The results of the output response of PP foam ferroelectret to pulse and the sinusoidal input signal are tabulated in Table 2. The efficiency of the energy harvesting performance of the PP ferroelectret is given by the number of output voltage generated by the ferroelectret sample divided by the input acceleration(g) supplied. As shown in Table 2, the voltage output generated from sinusoidal input is almost two times more than that from pulse input. Hence the output efficiency per unit acceleration for a pulse and sinusoidal input is 1.513 V/g and 2.837 V/g respectively. Since output generated from sinusoidal input is 47%

more efficient compared to that produced from pulse input, it can be said that sinusoidal input is more suitable for the energy harvesting applications of PP ferroelectret material. This result is reasonable because the sine waveform has the least harmonics and retains its wave shape when added to another sine wave of the same frequency and phase. So, it does not have much loss in its value compared to other shapes of waveforms.

**Table 2.** Pulse and sinusoidal input-output comparison for the PP sample.

| Waveform | Acceleration (g) | Voltage Generated (V) | Efficiency (V/g) |
|---|---|---|---|
| Pulse | 0.42 | 0.64 | 1.51 |
| Sinusoidal | 0.42 | 1.2 | 2.84 |

In d33 mode, the ferroelectret material is mechanically compressed in a direction perpendicular to the electrodes on its surfaces. This compression generates a charge imbalance and results in a voltage output across the electrodes. When a sinusoidal waveform is applied, it causes the piezoelectric material to vibrate at its fundamental resonance frequency, which maximizes the output voltage.

On the other hand, pulse excitation, such as a square or rectangular waveform, can excite multiple resonance modes simultaneously. This can result in a lower overall output voltage because the energy is distributed across multiple modes instead of being concentrated on the fundamental mode. Another factor that can affect the output voltage is the damping in the system. Pulse excitation can lead to more damping and energy losses compared to sinusoidal excitation, which can further reduce the output voltage.

## 5. Result of Frequency Response

Different weights of seismic masses should be applied while testing d33 mode ferroelectret energy harvesters to explore the optimal conditions for energy harvesting. The weight of the seismic mass affects the mechanical damping of the system, which can have a significant impact on the output power and efficiency of the energy harvester. In d33 mode, the piezoelectric material is mechanically compressed in a direction perpendicular to the electrodes on its surfaces. This compression generates a charge imbalance and results in a voltage output across the electrodes. However, the amount of mechanical deformation that can be applied to the piezoelectric material is limited by the mechanical damping of the system, which can reduce the output power and efficiency.

By applying different weights of seismic masses, it is possible to adjust the mechanical damping of the system and explore the optimal conditions for energy harvesting. A higher weight of seismic mass can increase the mechanical damping and reduce the mechanical deformation of the ferroelectret material, which can lower the output power and efficiency. On the other hand, a lower weight of seismic mass can reduce mechanical damping and allow for a higher degree of mechanical deformation, which can increase output power and efficiency. Therefore, by systematically varying the weight of the seismic mass, it is possible to identify the optimal conditions for energy harvesting, which can maximize the output power and efficiency of the d33 mode ferroelectret energy harvester. The seismic mass is typically a heavy metal or ceramic material that is attached to the piezoelectric material using a mechanical coupling, in our case we use a metal seismic mass of a minimum of 50 g and a maximum of 150 g. The average seismic mass weight used in piezoelectric energy harvester setups in d33 mode can vary depending on the specific application and the properties of the piezoelectric material being used.

As shown in Figure 5 by adding or removing the washer plates the compressive force on the sample can be varied. The frequency response of the PP sample is evaluated for different applied weights. The average voltage generated of three trails for each seismic mass is shown in Figure 5. Acceleration applied is 0.4 g in all cases and load resistance is 300 KΩ.

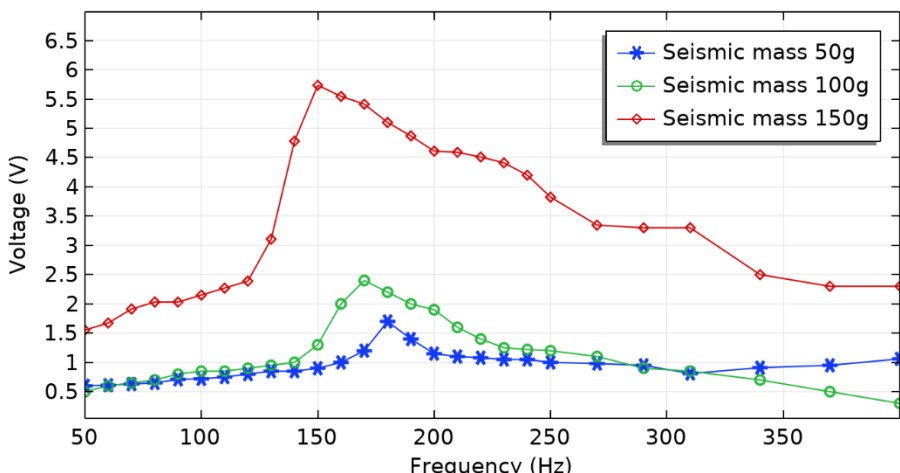

**Figure 5.** Frequency response of PP foam for 0.4 g acceleration and different applied weights.

As the applied weight increases the voltage generated also increases. The resonant frequency decreases as the applied weight increases, as seen in the Figure above. All the graphs show that the samples have wide operating bandwidths, are not suitable for low-frequency applications, and need at least 150 Hz to show substantial voltage results. Figure 6 shows the power generated versus frequency. At resonant frequency the power generated by the sample for a 50 g weight is around 10 µW, for a 100-g weight power is 20 µW, and for a 150-g weight power generated is 110 µW.

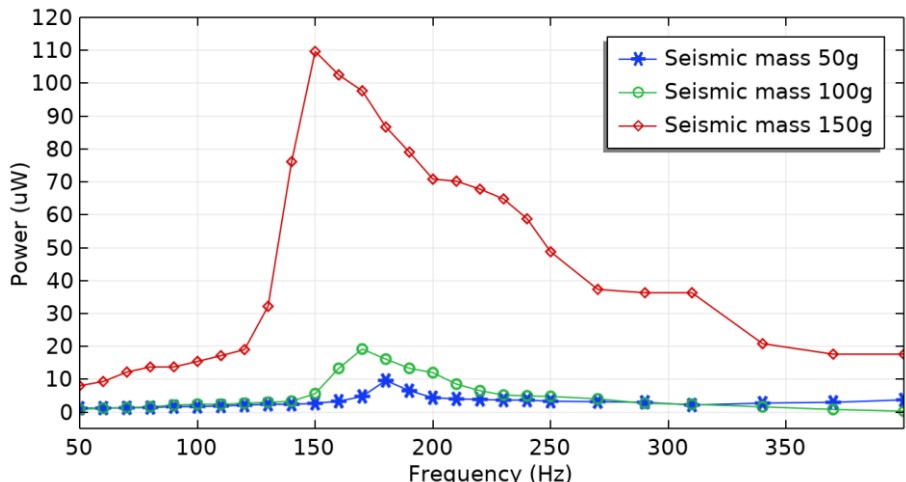

**Figure 6.** Power generated by PP foam for 0.4 g acceleration and different applied weights.

## 6. Discussion

The voltage output generated from sinusoidal input is almost two times more than that from pulse input. Hence the output efficiency per unit acceleration for a pulse and sinusoidal input is 1.513 V/g and 2.837 V/g respectively. Since output generated from sinusoidal input is 47% more efficient compared to that produced from pulse input, it can be said that sinusoidal input is more suitable for the energy harvesting applications of PP ferroelectret material. This result is reasonable because the sine waveform has the least harmonics and retains its wave shape when added to another sine wave of the same frequency and phase. So, it does not have much loss in its value compared to other shapes of waveforms.

So one can conclude that the response of the ferroelectret material to a given mechanical stress is not linear and depends on various factors, including the excitation waveform, amplitude, and frequency of the input vibration. When a sinusoidal waveform is applied

to a ferroelectret material, the material generates an electrical charge that is proportional to the amplitude and frequency of the input vibration. However, when a complex waveform is applied to the material, such as a square wave or a pulse wave, the material exhibits a nonlinear response due to its hysteresis and other inherent material properties.

Moreover, the behavior of the ferroelectret material also depends on the direction of the applied mechanical stress relative to the polarization direction of the material. In the d33 mode of ferroelectret energy generation, the material is subjected to a mechanical stress perpendicular to its polarization direction, which maximizes its d33 piezoelectric response. However, for other modes of vibration, such as d31 or d32, the response of the material is different, and the energy conversion efficiency may be lower. Therefore, the behavior of the ferroelectret material is different for different kinds of waveforms of input vibration frequency, and it is essential to carefully consider the material properties and the response of the material to external stimuli when designing piezoelectric energy harvesters for specific applications.

Concerning the frequency response, the resonant frequency of a ferroelectret energy harvester depends on the mechanical properties of the ferroelectret material and the mass of the seismic mass. The resonant frequency is determined by the natural frequency of the piezoelectric material and the seismic mass when they are mechanically coupled together.

In ferroelectret energy harvesters, the ferroelectret foam material has a high stiffness, which means that it can vibrate at a higher frequency than softer materials. Additionally, ferroelectret foam materials have a low mass, which means that the seismic mass needed to mechanically couple the piezoelectric material to an external vibration source can also be relatively low. This combination of high stiffness and low mass results in a higher resonant frequency for polypropylene foam is found to be around 150 Hz. However, the resonant frequency of a ferroelectret energy harvester can be influenced by various factors, including the size and shape of the ferroelectret material and seismic mass, the electrical circuitry used to extract power from the device, and the environmental conditions in which the device operates. Therefore, it is important to carefully design and optimize the energy harvester for the specific application and operating conditions to maximize its power output and efficiency.

It is observed that the resonant frequency of the sample reduces as the weight of seismic mass increases. Resonant frequency for 50 g, 100 g, and 150 g is 180 Hz, 170 Hz, and 150 Hz respectively. At the resonant frequency, the power generated by the sample for a 50-g seismic weight is around 10 μW, for 100-g weight power is 20 μW, and for a 150-g weight power generated is 110 μW. This voltage generated depends mainly upon the material's piezoelectric coupling, internal, and external factors such as internal resistance and ambient temperature, etc. A study of the influence of such factors is crucial as the energy in environmental vibrations is inherently low. Compared to conventional piezoelectric polymers, these ferroelectrets have more than ten times higher piezoelectric coefficients. Hence it becomes interesting to investigate the harvesting performance of ferroelectrets. This substance can multiply the energy harvester power generation capacity many folds such that a highly efficient and economically viable energy harvesting system is built whose application area is widened along with easy installation. The results of the frequency response for each of the different seismic mass conditions is shown in Table 3 below.

**Table 3.** Measured data of PP sample.

| Seismic Mass (g) | Acc (g) | Resonant Frequency (Hz) | Rms Voltage (V) | Power (μW) | Power Density (μW/g/mm$^3$) |
|---|---|---|---|---|---|
| 50 | 0.4 | 180 | 1.7 | 9.6 | 3.4 |
| 100 | 0.4 | 170 | 2.4 | 19.2 | 6.8 |
| 150 | 0.4 | 150 | 5.7 | 109.7 | 39.1 |

Ferroelectret energy harvester is producing power of 39 ($\mu$W/g/mm$^3$) for 150 g of seismic mass which is more than other similar devices mentioned in the literature. The possible factors that may be contributing to the improved performance are the design and fabrication of the harvester structure, operating conditions, quality of the ferroelectret material, and calibration of measurement equipment. The conditions under which the harvester is being operated may be more favorable for power generation. This could include factors such as the shape of the input excitation waveform, frequency and amplitude of the input vibration, and load impedance. The quality and properties of the ferroelectret material itself can vary depending on factors such as the manufacturing process, the quality of the raw materials, and the conditions of storage and handling. If your ferroelectret material has a higher density of charges or more uniform porosity, it may be more efficient at converting mechanical energy into electrical energy. All the literature review and the experiments conducted in this study suggest that it is also worth noting that the performance of piezoelectric energy harvesters can be highly dependent on the specific operating conditions and environmental factors. Therefore, it is important to conduct a thorough characterization of the harvester under different conditions to fully understand its performance and potential applications.

However, there are some shortcomings also that need to be addressed. The ferroelectret could have a slight static deflection due to the seismic mass and is also dynamically deflected by the seismic mass in response to acceleration from an electrodynamic shaker. It is worth noting that the use of seismic masses can introduce some challenges in the design and construction of piezoelectric energy harvesters, as they add additional weight and complexity to the system. Ferroelectret energy harvesters typically have higher resonant frequencies around 150 Hz depending on the amount of seismic mass, because the ferroelectret foam material used in these devices has a relatively high stiffness and low mass, which results in a higher resonant frequency. This feature could limit some potential applications of these devices unless more methods are researched to lower the resonant frequency.

## 7. Conclusions

In the experiment, small seismic masses of 50 g, 100 g, and 150 g were used for the purpose to test the energy harvesting capacity of ferrolelectrets. The results showed that the energy harvesting capacity of ferrolelectrets is dependent on the size of the seismic mass, as the output voltage and power density increase with increasing seismic mass. To justify the design approach of this energy harvester, we could emphasize the following points:

a.  The voltage output generated from sinusoidal input is almost two times more than that from pulse input. Hence the output efficiency per unit acceleration for a pulse and sinusoidal input is 1.513 V/g and 2.837 V/g, respectively. Since output generated from sinusoidal input is 47% more efficient compared to that produced from pulse input, it can be said that sinusoidal input is more suitable for the energy harvesting applications of PP ferroelectret material.

b.  The device achieves a power density output of 39 ($\mu$W/g/mm$^3$), which is ten times higher than that of PVDF (a commonly used material for energy harvesting). This high-power density indicates that the energy harvester is efficient in converting vibrations into electrical energy.

c.  The design demonstrates improved efficiency with larger seismic masses. This scalability feature allows for flexibility in choosing the appropriate seismic mass for specific applications, optimizing the device's performance according to the available mass.

d.  The proposed energy harvester has a resonant frequency that can be tuned within the range of 150 to 200 Hz. This broad frequency range allows for potential applications in various environments where different vibration frequencies may be present.

e.  The energy harvester can still generate comparable power output to PVDF even with small seismic masses. This feature enables the device to be miniaturized and integrated into compact electronic systems or wearable devices where space is limited.

For further research, the focus is to find and incorporate new kinds of ferroelectrets in this harvester device, which have a higher piezoelectric coefficient. The goal now is to reach further within the prototype and be able to assess the harvester on other piezoelectric materials so that we can increase harvester efficiency. Piezoelectricity may give us the chance to harvest the energy found within all moving objects to provide a sustainable and cleaner future.

**Author Contributions:** C.R. and V.M. jointly conceived the idea. C.R. and V.M. designed and fabricated the device, built the experimental setup and performed experiments. C.R. wrote the manuscript with contributions from all co-authors. All authors have read and agreed to the published version of the manuscript.

**Funding:** This research received no external funding.

**Data Availability Statement:** The data presented in this study are available on request from the corresponding author.

**Conflicts of Interest:** The authors declare no conflict of interest.

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
