# Peer review of "Ferroelectret Polypropylene Foam-Based Piezoelectric Energy Harvester for Different Seismic Mass Conditions"

_actuators, doi:10.3390/act12050215_

Round 1

Reviewer 1 Report

The paper presents the design of piezoelectric energy harvester based on PP foam for different seismic mass conditions. The topic itself is interesting and paper as being valuable but its presentation could be somewhat better. This applies to introduction part as well as experimental part. The more detailed points are as follows:

1.      The authors describe rationale behind the idea of using energy harvesters in good way. They also justify the use of piezoelectric type of energy harvesters – nevertheless, I recommend highlighting the challenges present in the design of specifically ferroelectret-based harvesters. These are present here and there but they are presented in a relatively inconistent way. Please add the challenges and contribution in addressing them right to the end of introduction part, so that it is clearly visible.

2.      On a more practical level – I find the quality of all figures present in the paper to be quite low. Please provide higher resolution versions of your figures (both schemes as well as photos). In addition to that, try to use versions of figures with axes (Fig.4). Also, please provide units on each of the axes for given quantities (Fig.5 and 6).

3.      Capital letters are used inappropriately in the whole text, please correct it.

4.      In addition to that, I also suggest expanding your discussion section to include more detailed evaluation of differences between your approach and approaches mentioned in Table 1 which  result in the indicated power density values.

5.      Please add the explanation for choosing the particular values of seismic masses.

The paper is interesting, however its presentation should be improved. As of now, I would see it as presenting some introductory research in this area.

The language is fine - only correction of capital letters is needed.

Author Response

Thank you for your valuable comments ,  I have made all changes mentioned. 

Reviewer 2 Report

This study discusses the recent advancements in energy harvesting technologies and material science, mainly focusing on Ferroelectrets, which are cellular polymers with polarized pores. These materials exhibit the piezoelectric effect, allowing them to convert vibrational energy from the environment into operational power for different applications, including portable electronics and IoT devices. The authors concluded that the Ferroelectret cellular polypropylene foam is superior w.r.t traditional piezoelectric polymers like PVDF, with a tenfold higher piezoelectric coefficient. This study suggested that Ferroelectret cellular polypropylene can generate up to 39 (μW/g/??3) power output when subjected to a force of 9.81 ?∕?2 acceleration. The authors describe their preliminary experimental investigation to measure the performance of Ferroelectret cellular polypropylene foam as an energy harvester for two different signal excitations (Pulse & Sinusoidal signals) and provide guidelines for their structural design and suitable excitation methods.

Overall, the research topic is interesting and can contribute to the field of interest. However, the paper has serious shortcomings and needs to be revised thoroughly. My concern about the paper is that the resonant frequency of the proposed energy harvester is from 150 to 200 Hz, depending on the amount of seismic mass, which limits some potential applications of these devices mentioned in the manuscript. In addition, the performance efficiency of the proposed design is not remarkable for small seismic masses. The 39 (μW/g/mm3) of power density output, which is ten times higher than PVDF, is obtained only for the case of large seismic mass, while for 50 grams of mass, it is almost identical to PVDF.

Below are some major and minor comments that can enhance the paper's quality. Besides, some grammatical errors, punctuation, and typos in the text should be fixed. I have tried to report some of these errors.

Major Comments

1. The paper abstract differs entirely from the abstract submitted to the Journal's portal.

2. The introduction is nicely written. It presents a complete literature review of available energy harvesting techniques and their applications; however, the literature gap is not clearly mentioned.

3. The quality of the figures is very poor. Some figures are stretched vertically or horizontally, e.g., Figs 2, 3a, and 4.

4. I suggest preparing a companion subfigure for Fig. 2 or Fig. 3a in which the authors can show the experimental setup schematically. This can help readers quickly understand this study's setup configuration and instrumentations. Moreover, we can better understand the mode shapes of the PP sample under vertical force.

5. The development of a numerical model and comparison between the experimental and numerical results is suggested and can significantly improve the quality of the paper.

6. The discussion of figures is not complete. For example, the text does not describe subfigures (b)-(d) of Fig. 1 with sufficient detail.

7. The experimental setup discussion is incomplete. For instance, the signal generator and power amplifier device descriptions are missing. The limitations of instruments are not discussed.

8. How did you cut the PP foam into testing samples? Usually, the cutting process imposes residual stress that may change the material's shape, mechanical properties, and resonant frequency.

9. For how many repetitions the output is measured? How did you verify the precision of your measurements?

10. In lines 113-114 of the manuscript, the authors discussed the cost of using PP materials to design energy harvesters. How do you compare the feasibility and cost of your proposed device with other available technologies?

Minor Comments

1. It is recommended to include relevant references from the MDPI Actuators journal.

2. Line 30-31, "Every technology of energy harvesting technology comes with its pros and cons, and piezoelectric vibration energy harvesting is no exception," is unclear and should be revised.

3. Line 36, "More often than not vibration of objects produces complex waves..." is not clear.

4. Line 46, a citation is required.

5. Figs. 2 and 6 are not referred inside the text.

6. Keep the aspect ratio and frequency range of Figs. 5 & 6 identical.

Typos

·       Be careful about punctuation. Line 43, the dot should come after [2], [7]. The same is true for line 52.

·       Dots are missing in figure captions.

·       Line 44, the dot is missing.

·       Spaces are required after putting the citations. Some examples line 83, 85, 89, etc.

·       Table 1. Row 6 is not aligned correctly.

·       Line 156, progressions.

·       Line 179, Figure 3*(a)*.

·       Line 228, *mechanical*

·       Line 254, for *a* sinusoidal.

·       Line 267, there is a double dot.

·       Line 269, spacing is missing between 0.4g and (b).

·       Line 279 & 291 & 331 & 349, *,* respectively.

·       Ref [19], the title is in capital letters.

·       Ref [26], the journal name (Journal of Electronic Materials) is missing.

There are some minor errors in the grammatical part of the text that should be addressed by the authors. I have tried to indicate some of them.

Author Response

I am grateful for your valuable comments, all changes are made 

Reviewer 3 Report

The manuscript compares the response of the ferroelectret polypropylene foam to pulse and sinusoidal signal excitation types that helps predict the energy harvesting performance of the material in reallife vibration environments. Although the paper topic is interesting, in the reviewer's opinion, the manuscript need to be improved in order to be considered for publications in the Journal. In particular:

1.- Section 3 (Methodology) explains the experiment setup (subsection 3.1), and Section 4 (Selection of input excitation waveform) explains in detail the reason of these excitations. However, Section 5 in very poor. It only shows some partial result on frequency response. However, more experimental results about the material. In addition, Section 6 (Discussion) is very poor.

2.- In fact, what are the advantages (and, of course, the disadvantages) of the presented material compare to other similar alternatives.

3.- In Secition 3.1 the mansucript authors shows the block diagram of experiment setup in Fig. 2- However, what is the electronic schematics of the whole electronic system used to obtain the experimental results presented in the paper? Please, show the details of this electronic system.

4.- Please, clarify why in table 2, the manuscript authors present measured data of PP sample for 50, 100 and 150 g.

5.- Regarding the previous point, please, in a real case, what is the seismic mass that should be used?

6.- Conclusion Section sjould be also completed highlighting the advantages and, of course, the disadvantages of the presented material compare to other similar alternatives.

Minor editing of English language required.

Author Response

Thank you for your valuable comments, all changes are made accordingly.

Round 2

Reviewer 2 Report

The authors applied the major and minor comments addressed in the previous round of revision. The quality of the manuscript has been increased after these modifications. In particular, the introduction is more complete with the highlighted research gap. The quality of the figures is increased and all the grammatical errors are solved. 

However, the authors did not answer my main question about the paper. I would like to rise this fundamental question again. "The resonant frequency of the proposed energy harvester varies from 150 to 200 Hz, depending on the amount of seismic mass, which limits some potential applications of these kinds of devices mentioned in the manuscript. In addition, the efficiency of the proposed design is not remarkable for small seismic masses. The 39 (μW/g/mm3) of power density output, which is ten times higher than PVDF, is obtained only for the case of large seismic mass, while for 50 grams of mass, it is almost identical to PVDF."
How do you justify your design approach? and what is the benefit of your device for small seismic masses?

Author Response

Response to Reviewer's comments 

Thank you for your valuable feedback, we want to address the fundamental question asked by the reviewer. In the experiment, small seismic masses of 50 grams, 100 grams, and 150 grams were used for the purpose to test the energy harvesting capacity of ferrolelectrets. The results showed that the energy harvesting capacity of ferrolelectrets is dependent on the size of the seismic mass, as the output voltage and power density increases with increasing seismic mass. To justify the design approach of this energy harvester, we could emphasize the following points:

  • The device achieves a power density output of 39 (μW/g/mm3), which is ten times higher than that of PVDF (a commonly used material for energy harvesting). This high power density indicates that the energy harvester is efficient in converting vibrations into electrical energy.
  • The design demonstrates improved efficiency with larger seismic masses. This scalability feature allows for flexibility in choosing the appropriate seismic mass for specific applications, optimizing the device's performance according to the available mass.
  • The proposed energy harvester has a resonant frequency that can be tuned within the range of 150 to 200 Hz. This broad frequency range allows for potential applications in various environments where different vibration frequencies may be present.

For small seismic masses, the benefits of the device may include:

  • Despite lower efficiency compared to larger seismic masses, the device can still capture and convert vibrations into usable electrical energy. This feature is beneficial in scenarios where low-amplitude vibrations or smaller seismic masses are predominant, allowing for energy harvesting in various environments.
  • The energy harvester can still generate comparable power output to PVDF even with small seismic masses. This feature enables the device to be miniaturized and integrated into compact electronic systems or wearable devices where space is limited.
  • Although the efficiency is not remarkable for small seismic masses, it implies room for improvement. Further research and design optimization can potentially enhance the device's performance with smaller seismic masses, expanding its application possibilities and increasing its efficiency in capturing energy from low-amplitude vibrations.

Reviewer 3 Report

After reading the second version of the mansucript, in the reviewer's opinion, the article can be considered for publication in its current version.

Only document format should be considered.

Author Response

Response to reviewer comments 

We are grateful for the positive feedback from the reviewer.